# Fungal Treatment for the Valorization of Technical Soda Lignin

**DOI:** 10.3390/jof7010039

**Published:** 2021-01-09

**Authors:** Mariane Daou, Clementina Farfan Soto, Amel Majira, Laurent Cézard, Betty Cottyn, Florian Pion, David Navarro, Lydie Oliveira Correia, Elodie Drula, Eric Record, Sana Raouche, Stéphanie Baumberger, Craig B. Faulds

**Affiliations:** 1French National Research Institute for Agriculture, Food and Environment (INRAE), UMR1163, Biodiversité et Biotechnologie Fongiques, Aix Marseille University, 13288 Marseille, France; mariane.daou@ku.ac.ae (M.D.); cmfs7383@gmail.com (C.F.S.); david.navarro@inrae.fr (D.N.); elodie.drula@inrae.fr (E.D.); eric.record@inrae.fr (E.R.); sana.raouche@univ-amu.fr (S.R.); 2Institut Jean-Pierre Bourgin, INRAE, AgroParisTech, Université Paris-Saclay, 78000 Versailles, France; amel.majira@inrae.fr (A.M.); laurent.cezard@inrae.fr (L.C.); betty.cottyn@inra.fr (B.C.); florian.pion@inrae.fr (F.P.); stephanie.baumberger@inrae.fr (S.B.); 3The Fungal Biodiversity and Biotechnology Laboratory, Centre International de Ressources Microbiennes—Champignons Filamenteux, INRAE, Aix Marseille University, 13288 Marseille, France; 4Plateforme d’Analyse Protéomique de Paris Sud-Ouest, INRAE, AgroParisTech, Université Paris-Saclay, Micalis Institute, 78350 Jouy-en-Josas, France; lydie.oliveira-correia@inrae.fr

**Keywords:** filamentous fungi, technical lignin, oxidoreductases, secretomic analysis, *Polyporus brumalis*, *Pycnoporus sanguineus*, *Leiotrametes menziesii*

## Abstract

Technical lignins produced as a by-product in biorefinery processes represent a potential source of renewable carbon. In consideration of the possibilities of the industrial transformation of this substrate into various valuable bio-based molecules, the biological deconstruction of a technical soda lignin by filamentous fungi was investigated. The ability of three basidiomycetes (*Polyporus brumalis*, *Pycnoporus sanguineus* and *Leiotrametes menziesii*) to modify this material, the resultant structural and chemical changes, and the secreted proteins during growth on this substrate were investigated. The three fungi could grow on the technical lignin alone, and the growth rate increased when the media were supplemented with glucose or maltose. The proteomic analysis of the culture supernatants after three days of growth revealed the secretion of numerous Carbohydrate-Active Enzymes (CAZymes). The secretomic profiles varied widely between the strains and the presence of technical lignin alone triggered the early secretion of many lignin-acting oxidoreductases. The secretomes were notably rich in glycoside hydrolases and H_2_O_2_-producing auxiliary activity enzymes with copper radical oxidases being induced on lignin for all strains. The lignin treatment by fungi modified both the soluble and insoluble lignin fractions. A significant decrease in the amount of soluble higher molar mass compounds was observed in the case of *P. sanguineus*. This strain was also responsible for the modification of the lower molar mass compounds of the lignin insoluble fraction and a 40% decrease in the thioacidolysis yield. The similarity in the activities of *P. sanguineus* and *P. brumalis* in modifying the functional groups of the technical lignin were observed, the results suggest that the lignin has undergone structural changes, or at least changes in its composition, and pave the route for the utilization of filamentous fungi to functionalize technical lignins and produce the enzymes of interest for biorefinery applications.

## 1. Introduction

As fossil fuel reserves continue to diminish, the development of sustainable sources of platform chemicals has included the investigation of the complete and systematic utilization of renewable plant biomass. Lignocellulosic biomass is rich in polysaccharides and (poly)aromatic compounds, which can be potentially converted into such chemicals deemed suitable to replace those coming currently from fossil fuels [1]. Moreover, they can be retrieved at a relatively low cost from forestry and agricultural residues, thus avoiding direct competition with food supply and land use issues. A current bottleneck in the use of such feedstock in industrial processes is the presence of substantial amounts of recalcitrant lignin (10–30%), resisting a wide range of chemical and biological attack and hindering the access of enzymes to cellulose, the main fermentable sugar stock [2]. To overcome this impasse, different pre-treatments have been integrated into lignocellulose biorefinery processes, including the delignification pre-treatments that generate technical lignins as by-products [3].

With a global annual production by the pulp and paper industry exceeding 100 million tons [4], mainly as Kraft lignins, lignosulfonates and soda lignins, technical lignins represent a potential source of renewable carbon. Chemical and thermochemical routes have already been shown to be efficient for lignin engineering [5], however, most of them are energy-consuming and catalytic conversions often require toxic and expensive chemicals. Today, 95% of the produced technical lignins are burnt as an energy source and the remaining 5% utilized in the production of dispersants, adhesives, and surfactants, or as antioxidants in plastics and rubbers [6]. To improve economic growth and durability for lignocellulose biorefineries, a more efficient valorization of this by-product through depolymerization and/or functionalization into higher value products is required [7].

In nature, the complexity and rigidity of lignin present in the secondary cell wall of vascular plants is mirrored by the advanced degradation mechanisms developed by lignocellulose-modifying microorganisms, in particular wood-decaying fungi. Previous studies have examined representatives of these fungi during growth on whole lignocellulosic biomass, measured the mineralization of labelled lignin from plants and mapped the secretion of many carbohydrate- and lignin-active enzyme activities [8,9,10]. A diverse repertoire of extracellular lignocellulolytic enzymes were identified and characterized during these studies (see [11] for review).

However, while most of the studies used whole biomass, native lignins and lignin-like monomers and oligomers as substrates, a limited number of data are available on the ability of fungi to modify technical lignin polymers. Native lignins undergo substantial structural changes after physical and chemical treatments resulting in the material referred to as technical lignins, which are chemically very different and should be considered as a distinct substrate [12,13]. Indeed, technical lignins are typified by a lower proportion of β-O-4 bonds (up to five times lower than native lignins) and the presence of neo-formed carbon–carbon bonds, together with new functional groups resulting from demethylation (e.g., catechol) or oxidation (e.g., Cα carbonyls) reactions [14]. Moreover, consequently to the partial depolymerization of the native polymers during delignification, technical lignins generally contain a few percent of low molar mass phenolics, including monomers. The *Polyporales* order of Basidiomycetes contains most of the known wood-rot fungi, where lignin degradation is linked to the secretion of organic acids, secondary metabolites and oxidoreductive enzymes [15]. Previous studies performed in our lab have identified three basidiomycetes (*Polyporus brumalis, Pycnoporus sanguineus* and *Leiotrametes menziesii*) exhibiting the selective delignification of wheat straw [16,17,18]. The strains were therefore considered as good candidates for the modification of wheat straw-derived technical soda lignins in this study. The response of the three white-rot fungi to a chemically heterogeneous commercial technical lignin (grass soda lignin Protobind 1000, GreenValue Enterprises LLC, Upper Providence Township, PA, USA) was investigated. This substrate is considered as a reference sample due to its availability and wide application in previous studies [13,19,20,21].

Enzymes secreted over three different periods of growth on the technical soda lignins were identified, bringing insights on the enzymatic degradation strategies of the studied white-rot fungi. The structural changes on the lignin substrates were deciphered to assess the possibility offered by fungal treatments to functionalize technical lignins.

## 2. Materials and Methods

### 2.1. Fungal Strains and Soda Lignin

Dikaryotic strains of the fungi *Polyporus brumalis* BRFM 985 (Pbr985), *Pycnoporus sanguineus* BRFM 1264 (Psan1264), and *Leiotrametes menziesii* BRFM 1369 (Lme1369) were obtained from the International Center of Microbial Resources, Marseille, France (CIRM-CF; https://www6.inra.fr/cirm_eng). All strains were previously identified by the morphological and molecular analysis of Internal Transcribed Spacer sequences using the expert database Fungene-db (http://www.fungene-db.org) or Genbank [22,23,24,25]. Fungal strains were maintained on Malt extract (Duchefa Biochemie, Haarlem, Netherlands) slants at 4 °C.

The soda technical lignin (Protobind 1000) was produced from a wheat straw and Sarkanda grass mix and purchased from GreenValue Enterprises LLC. The sample was analyzed for Klason lignin (88.1%), carbohydrates (1.9%, of which 1.2% xylose, 0.3% arabinose, 0.1% galactose and 0.2% glucose), free phenolic monomers (1.4%) and ash (1.4%) contents. Klason lignin content was determined gravimetrically as described by Dence [26]. Carbohydrates were analyzed according to Sipponen [27]. The extraction of free phenolics was performed as previously described [28]. All the analyses were performed in duplicate (error < 3%).

### 2.2. Culture Conditions

#### 2.2.1. Cultures on Lignin-Containing Plates

From agar slants, the basidiomycetes strains were transferred and grown on agar plates containing 20 g·L^−1^ malt extract and 1 g·L^−1^ yeast extract. Disc cuts of 0.5 cm from the growing mycelium edge were then used to inoculate minimal media plates containing 10 g·L^−1^ technical soda lignin. The minimal media contained 0.3 g·L^−1^ NaNO_3_, 0.026 g·L^−1^ KCl, 0.026 g·L^−1^ MgSO_4_·7H_2_O, 0.041 g·L^−1^ KHPO_4_, 0.00044 g·L^−1^ ZnSO_4_·7H_2_O, 0.00022 g·L^−1^ H_3_BO_3_, 0.0001 g·L^−1^ MnCl_2_·4H_2_O, 0.0001 g·L^−1^ FeSO_4_·7H_2_O, 0.000032 g·L^−1^ CoCl_2_·6H_2_O, 0.000032 g·L^−1^ CuSO_4_·6H_2_O, 0.000022 g·L^−1^ (NH_4_)_6_Mo_7_O_24_·4H_2_O, 0.001 g·L^−1^ Na_2_EDTA and 20 g·L^−1^ agar. Control plates were supplemented with 1 g·L^−1^ glucose. Fungal growth was followed daily over 14 days by measuring the radial expansion of the fungi (cm·day^−1^) and the surface mycelial density was observed. All cultures were performed in duplicate.

#### 2.2.2. Mycelia Preparation for Liquid Cultures

Five discs (0.5 cm) were cut from the growing mycelium edge on malt agar plates for each of the three strains, and used to inoculate Roux flasks containing 200 mL of medium (10 g·L^−1^ maltose, 0.5 g·L^−1^ yeast extract, 1.84 g·L^−1^ diammonium tartrate, 0.2 g·L^−1^ KH_2_PO_4_, 0.0132 g·L^−1^ CaCl_2_, 0.5 g·L^−1^ MgSO_4_, 2.5 g·L^−1^ thiamine, 0.074 g·L^−1^ FeSO_4_·7H_2_O, 0.0777 g·L^−1^ ZnSO_4_·7H_2_O, 0.0363 g·L^−1^ MnSO_4_·H_2_O and 0.0072 g·L^−1^ CuSO_5_·5H_2_O). The flasks were statically incubated at 30 °C for 15 days. The collected mycelium for each strain was then filtered, washed and homogenized in water (Ultra-Turrax 9500 rpm for 60 s) to have a final fresh mycelial concentration of 100 mg·mL^−1^. The dry biomass weights were determined after incubation for 24 h at 105 °C.

#### 2.2.3. Liquid Cultures on Lignin

Each 250 mL baffled Erlenmeyer flask containing 100 mL of media (10 g·L^−1^ technical soda lignin, 0.5 g·L^−1^ yeast extract, 1.842 g·L^−1^ diammonium tartrate, 0.0025 g·L^−1^ thiamine, 0.2 g·L^−1^ KH_2_PO_4_, 0.0132 g·L^−1^ CaCl_2_, 0.5 g·L^−1^ MgSO_4_, 0.074 g·L^−1^ FeSO_4_·7H_2_O, 0.0777 g·L^−1^ ZnSO_4_·7H_2_O, 0.0363 g·L^−1^ MnSO_4_·H_2_O and 0.0072 g·L^−1^ CuSO_5_·5H_2_O) was inoculated with 5 mL of the ground mycelium suspension, described above. Two control conditions were used. The first contained 20 g·L^−1^ maltose without lignin and the second was supplemented with 2.5 g·L^−1^ maltose in addition to lignin. Incubation was carried out in the dark at 30 °C in a rotary shaker at 120 rpm for 14 days. The flasks were prepared in duplicate for each strain and each condition.

Mycelia were harvested at days 3, 7 and 14 by filtering through Miracloth, washing with sterile water and gently squeezing to remove as much moisture as possible. The harvested material was then flash frozen in liquid nitrogen and stored at −80 °C until use. Culture supernatants containing secreted proteins were also collected, filtered through the polyethersulfone membrane (pore size 0.22 µm; Express Plus, Merck Millipore, Darmstadt, Germany), and diafiltered with 50 mM sodium acetate (pH 5.0) and concentrated using a Vivaspin polyethersulfone membrane with a 10 kDa cutoff (Sartorius, Goettingen, Germany). Concentrated samples (secretomes) were then stored at −20 °C until use. The flow through from the concentration step was also collected for analysis and stored at −20 °C.

### 2.3. DNA Extraction and Mycelial Dry Weight Determination on Lignin

Growth on the technical soda lignin PB1000 was followed by quantifying the fungal DNA material. Frozen mycelia were ground using a SamplePrep 6770 FreezerMill (SPEX® SamplePrep, Metuchen, NJ, USA). The biomass (20, 40, 60, 80 and 100 mg fresh weights) dry weights were determined after incubation for 24 h at 105 °C. This allowed the correlation between mycelial fresh and dry weights (Appendix A). Fungal DNA was extracted following the method previously described by Zhou [29]. NucleoSpin^®^ Plant II kit (Macherey-Nagel, Dueren, Germany) was used according to the manufacturer’s instructions except for the changes described by Zhou [29]. A NanoDrop 2000 spectrophotometer (Thermo Scientific, Waltham, MA, USA) was used to determine the concentration of the extracted DNA. The conversion factors between the extracted fungal genomic DNA from negative controls not containing lignin and mycelium fresh weights were established (Appendix A). The calculated conversion factors were then applied to determine the dry weight from the extracted DNA for the samples containing lignin. Dry weight determination and DNA extractions were performed in triplicates for each selected weight.

### 2.4. Enzyme Activity

Laccase and peroxidase activities were investigated in the supernatants of cultures on lignin. Laccase activity was estimated by following the oxidation of 0.5 mM 2,2′-azino-bis (3-ethylbenzothiazoline-6-sulphonic acid) (ABTS) in 50 mM tartrate buffer pH 4 at 420 nm [30]. Lignin peroxidase (LiP) activity was tested on 2 mM veratryl alcohol in 50 mM tartrate buffer pH 3 and in the presence of 0.4 mM H_2_O_2_ [31]. The oxidation of veratryl alcohol was followed spectrophotometrically at 310 nm. Manganese peroxidase (MnP) activity was measured by following the formation of the Mn^3+^-tartrate complex in 50 mM tartrate buffer pH 5 at 238 nm [32]. All reactions were carried out in duplicate, and adequate controls with boiled (10 min at 100 °C) culture supernatants were performed.

### 2.5. Proteomic Analysis

Short SDS-PAGE runs were performed allowing the extracellular proteins contained in the secretomes of cultures on technical soda lignin PB1000 (10 µg) to migrate 0.5 cm. The gels were stained with Coomassie blue and each lane was excised and sent to PAPPSO platform facilities (http://pappso.inra.fr/) for protein identification. Gel bands were washed with 10 mM dithiothreitol (Sigma-Aldrich, St. Louis, MO, USA) and 55 mM iodoacetamide (Sigma-Aldrich). In-gel tryptic digestion was performed overnight with 200 ng of trypsine (Promega, Madison, WI, USA) in 50 mM bicarbonate (Sigma-Aldrich) buffer at 37 °C. Peptides were extracted with 0.5% trifluoroacetic acid (TFA) in 50% acetonitrile (ACN), dried and re-suspended in 80 µL of 0.1% TFA in 2% ACN for analysis in high-resolution mass spectrometry.

Liquid chromatography–mass spectrometry (LC–MS/MS) analyses were performed using a NanoLC Ultra system (Eksigent, Dublin, CA, USA) connected to a Q-Exactive Plus mass spectrometer (Thermo Scientific) and an Ultimate 3000 RSLC system (Thermo Scientific) coupled to an LTQ-orbitrap discovery mass spectrometer (Thermo Scientific) by nanoelectrospray ion source on both systems. Parameters and conditions of the two used methods are available in the Appendix A”.

All MS/MS spectra were integrated against the JGI databases for *P. sanguineus* BRFM 1264 v1.0 (https://genome.jgi.doe.gov/Pycsa1/Pycsa1.home.html), *P. brumalis* BRFM 1820 v1.0 (https://genome.jgi.doe.gov/Polbr1/Polbr1.home.html), and *L. menziesii* CIRM-BRFM 1781 v1.0 (https://genome.jgi.doe.gov/Tramen1/Tramen1.home.html), using the X!TandemPipeline (X!Tandem version 3.4.3), the open search engine developed by PAPPSO (http://pappso.inra.fr/bioinfo/xtandempipeline/). Precursor mass tolerance was 10 ppm and fragment mass tolerance was 0.02 and 0.5 Da on the Q-Exactive Plus and LTQ-Orbitrap mass spectrometers, respectively. Data filtering was achieved according to a peptide *E*-value < 0.01, protein *E*-value < 10 × 10^−4^ and to a minimum of two identified peptides per protein. Some of the proteins were only detected by one of the used methods therefore identifications from both mass spectrometers were combined.

All amino acid sequences were obtained from JGI Mycocosm [33]. The secretory signal peptides were predicted using the SignalP 5 [34]. The amino acid sequences were aligned by Multiple Sequence Alignment using Clustal Omega tool from EMBL-EBI [35]. The phylogenetic analysis was performed using maximum likelihood using the Molecular Evolutionary Genetic Analysis software version X (MEGAX) [36].

### 2.6. Analysis of the Soluble Lignin Fraction

#### 2.6.1. Sample Preparation

Ten microliters (10 mL) of culture supernatant from day 14 of growth and the control (lignin dissolved in the culture media in the absence of fungi and incubated under the same conditions) were extracted with 10 mL ethyl acetate (EA) after pH adjustment to 3–4 (2 M HCl aqueous solution). The aqueous layer was further extracted by 2 × 20 mL EA. The combined EA extracts were dried over MgSO_4_ and concentrated under reduced pressure below 45 °C. The residues were analyzed by HPSEC and LC–MS, after dissolution in the proper solvents.

#### 2.6.2. HPSEC Analysis

Aliquots of the EA extraction residues were dissolved in tetrahydrofuran (THF; 1 mg·mL^−1^) and the solutions filtered (0.45 μm, polytetrafluoroethylene membrane, Gelman) before injection on a HPSEC (Dionex Ultimate 3000, Thermo Scientific) apparatus equipped with a styrene–divinylbenzene PL-gel column (5 µm, 100 Å, 600 mm × 7.5 mm I.D., Polymer Laboratories, Church Stretton, United Kingdom) and photodiode array (PDA) detector set at 280 nm and using THF (1 mL·min^−1^) as eluent. Degrees of polymerization assigned to the different zones of the chromatograms were assessed from the apparent molar masses determined by a calibration curve based on polyethylene oxide standards (Igepal, Sigma-Aldrich) and the injection of pure phenolic monomers and dimers [37]. Toluene was used as the internal standard (IS) to normalize the chromatograms with respect to retention time.

#### 2.6.3. LC–MS Analysis

Aliquots of the EA extraction residues were dissolved in ACN (1 mg·mL^−1^) and the solutions were filtered (0.45 µm, GHP Acrodisc, Pall Gelman, Port Washington, NY, USA) before injection on an ultra-high performance liquid chromatography (UHPLC; Thermo Scientific) apparatus combined with an electrospray ionization mass spectrometer (ESI)–MS and PDA co-detection. UHPLC analysis was performed using a C18 column (Highpurity, Thermo Electron Corporation, 2.7 µm, 50 mm × 2 mm I.D.mm, Millipore), a 5–100% vol. aqueous ACN, 1‰ HCOOH gradient (30 min), and 0.4 mL·min^−1^ flow rate. Negative ion ESI–MS spectra (120–2000 m/z) were acquired using a quadrupole-time of flight (Q-TOF) spectrometer (Impact II, Bruker, Billerica, MA, USA) setting the needle voltage at 4 kV and the desolvating capillary temperature at 350 °C. The peaks were assigned according to the mass of the deprotonated ions and fragmentation pattern, and to the theoretical masses expected from the different types of lignin oligomers. Assignments of the phenolic monomers were confirmed by the injection of pure commercial compounds.

### 2.7. Analysis of Insoluble Lignin Fraction

#### 2.7.1. HPSEC Analysis

Residual technical soda lignin PB1000 was recovered after the exposure to fungi by the centrifugation of the culture medium on day 14. The control consisted of lignin dissolved in the culture media in the absence of fungi and was incubated and recovered under the same conditions. The HPSEC analysis of the insoluble lignin fraction was performed as described for the lignin soluble fraction. The solubility of the samples in THF was of 90% and the compounds eluted in the void volume (peak at 11.5 min) were assigned to polymers of apparent molar masses above 5000 g·mol^−1^, according to the column specifications (Agilent Technologies, Santa Clara, CA, USA).

#### 2.7.2. Quantitative ^31^P NMR and Samples Preparation

Derivatization of the samples with 2-chloro-4,4′,5,5′-tetramethyl-1,3,2-dioxaphospholane (TMDP, Sigma-Aldrich) was performed according to a published procedure [38]. The derivatized samples (20 mg) were dissolved in 400 µL of a mixture of anhydrous pyridine and deuterated chloroform (1.6:1 *v/v*). Then, 150 µL of a solution containing cyclohexanol (6 mg·mL^−1^) and chromium(III)acetylacetonate (3.6 mg·mL^−1^) was added, which served as IS and a relaxation reagent, respectively, and 75 µL of TMDP. Nuclear magnetic resonance (NMR) spectra were acquired without proton decoupling in CDCl_3_ at 162 MHz, on a Bruker Ascend 400 MHz spectrometer. A total of 128 scans were acquired with a delay time of 6 s between two successive pulses. The spectra were processed using Topspin 3.1. All chemical shifts were reported in parts per million relative to the product of phosphorylated cyclohexanol (IS), which has been observed to give a doublet at 145.1 ppm. The content in the hydroxyl groups (in mmol·g^−1^) was calculated on the basis of the integration of the phosphorylated cyclohexanol signal and by the integration of the following spectral regions: aliphatic hydroxyls (150.8–146.4 ppm), condensed phenolic units (145.8–143.8 ppm; 142.2–140.2 ppm), syringyl phenolic hydroxyls (143.8–142.2 ppm), guaiacyl phenolic hydroxyls (140.2–138.2 ppm), *p*-hydroxyphenyl phenolic hydroxyls (138.2–137.0 ppm), and carboxylic acids (136.6–133.6 ppm).

#### 2.7.3. Thioacidolysis

The thioacidolysis of lignins (5 mg) was carried out according to [39], using heneicosane (C_21_H_44_, Sigma-Aldrich) as IS. Lignin-derived *p*-hydroxyphenyl (H), guaiacyl (G), and syringyl (S), the thioacidolysis monomers were analyzed as their trimethylsilyl derivatives by a gas chromatography−mass spectrometry (GC–MS) instrument (Saturn 2100, Varian, Palo Alto, CA, USA) equipped with a poly(dimethylsiloxane) capillary column (30 m × 0.25 mm; SPB-1, Supelco, Bellefonte, PA, USA) and using the following heating program: 40 to 180 °C at 30 °C min^−1^, then 180 to 260 °C at 2 °C min^−1^. The mass spectrometer was an ion trap with an ionization energy of 70 eV and positive mode detection. The determination of the thioethylated H, G, and S monomers was performed from ion chromatograms reconstructed at *m/z* 239, 269, and 299, respectively, as compared to the IS signal measured from the ion chromatogram reconstructed at *m/z* (57 + 71 + 85). The molar yield of the detected thioethylated monomers was expressed with respect to the total amount of sample submitted to thioacidolysis.

## 3. Results and Discussion

### 3.1. Growth in the Presence of Lignin

#### 3.1.1. Growth on Lignin-Containing Plates

The ability of the three basidiomycete strains Pbr985, Psan1264 and Lme1369 to grow on 10 g·L^−1^ soda lignin derived from wheat straw was investigated over a 14-day period. The soda lignin was selected as a substrate showing a Klason lignin content close to 90%, and ash and carbohydrate contents lower than 2%, which are common characteristics of technical lignins [13] and make it a substrate of interest to test the ability of fungi to modify the phenolic fraction of technical lignins. The phenolic fraction was composed of monomers (1.4%, mainly acetosyringone, vanillin, syringaldehyde, and *p*-coumaric acid), oligomers and polymers (Appendix A).

All tested strains could grow on lignin-containing plates at different rates (Figure 1A). Psan1264 and Pbr985 grew more rapidly than Lme1369 in the presence of lignin alone.

The mycelial density of all three strains was always observed to be low on lignin and lignin extracts, while moderate to abundant on lignin plates supplemented with glucose (Figure 1B). Branching density is mainly controlled by the nutritional and environmental conditions. Highly dense and branched mycelia require larger amounts of nutrients per unit area of growth environment [40]. Simple monosaccharides, like glucose in this case, are easily utilized by fungi compared to complex lignin molecules, which explains the observed differences in mycelial density. However, fungi with lower mycelial density will extend further in order to create an effective balance between the energy costs, resources, and transport adeptness and resistance to damage [40]. The strategy used by fungi to ensure this balance varies between strains and this could possibly be the factor responsible for the differences observed in our current study.

A slight orange pigmentation of the mycelia was observed in the case of Psan1264 when grown on technical lignin alone. Cinnabarinic acid, a phenoxazinone compound, is known to be responsible for the orange color associated with Pycnoporus species [41]. This molecule is produced from the laccase-catalyzed oxidation of the precursor 3-hydroxyanthranilic acid [42]. Laccases also catalyze the oxidation of a wide range of phenolic compounds and non-phenolic compounds in the presence of mediators [43] and are therefore known to be involved in lignin degradation [44]. Consequently, the observed orange color on lignin-containing plates suggests that the production of laccases is induced by growth on the technical lignin.

#### 3.1.2. Liquid Cultures and Enzyme Activities

Fungi behave differently when they are grown on solid or liquid medium [45]. Therefore, the ability of the three strains to grow in liquid cultures containing the technical soda lignin was assessed. As the presence of insoluble lignin in the cultures would interfere with the measured dry weight, growth was followed by extracting and quantifying fungal genomic DNA. Furthermore, using this method allowed for the quantification of fungal mycelium remaining attached to the solid substrate.

The determination of the mycelia produced showed that, although growth was more significant when maltose was supplied, the three strains could grow in medium containing technical lignin in the absence of maltose (Figure 2). The log phase of fungal growth occurred between days 0 and 3 for all strains and conditions except for Pbr985 where the maximum mycelial mass was measured at day 7 of culture. The highest mycelial production was measured with Pbr985, which additionally showed similar growth at day 3 in the presence and absence of maltose. This suggests that this substrate might act as source of energy and carbon for wood-degrading fungi. Indeed, the technical lignin used in the present study contained 1.9% wt. carbohydrates, most probably oligomeric units remaining strongly bound to the lignin after the alkaline processing of the original biomass. However, the fine content of these carbohydrates in the lignin-containing cultures (0.0285% wt.) was too low for only the carbohydrate to account for the observed growth. The ability of some soil microorganisms to live on different types of natural and technical lignins was previously reported [46,47,48].

In all cases, a decrease in mycelial matter was observed after maximum growth was reached. This could be explained by the limitation of exponential growth as the mycelia expand in size, due to the restricted nutrient diffusion to the central cells [49]. Living cells produce toxic compounds, including reactive oxygen species, which would inhibit the growth of the peripheral mycelia. The production of these toxins and the impairment of the cellular functions result in molecular damage and cell death.

The in vitro oxidation of ABTS by culture supernatants varied between the strains. ABTS oxidation can be indicative of laccase activity in these supernatants, although other enzymes in the culture supernatant and generated free radicals could also result in the oxidation of ABTS. For all three strains, Pbr985, Psan1264 and Lme1369, ABTS oxidation was highest when both lignin and maltose were present in the media (Table 1) suggesting that while maltose ensured enhanced fungal development at early stages of growth, the technical lignin acted directly as a laccase inducer. Laccases are known to be produced constitutively by *P. cinabarinus* and *P. sanguineus* during primary metabolism [50,51]. This was observed during growth on maltose alone for Psan1264 and Pbr985 at days 3 and 7 of growth. However, ABTS oxidation was more significant at day 14 in lignin-containing conditions for the three basidiomycetes, which supports the laccase-inducing effect of this substrate over time. The highest ABTS oxidation was detected with Psan1264 (3175.8 nkat·mg^−1^ dry mycelia at day 14), a fungus known to be a high thermostable laccase producer, especially in the presence of woody material [52].

While no measurable MnP activity was detected for any of the strains, potential LiP-type activity was detected and showed strong variations (Table 1). Interestingly, veratryl alcohol oxidation was detected in the culture supernatant of Pbr985, which does not have any annotated LiP in its genome [18]. The detected activity could be due to either uncharacterized peroxidase in the secretome or other enzymes acting on veratryl alcohol as a substrate. The oxidation of veratryl alcohol was significantly higher in the presence of lignin compared to the maltose control condition and reached its maximum between days 7 and 14 of growth.

In view of these results, proteomic analyses were carried out to obtain further insight into the secreted enzymes during growth on technical lignin, supplemented or not with maltose.

### 3.2. Proteomic Analysis

Proteomic analysis was performed on the secretomes of the fungi at the three different time periods of growth on technical lignin. The three basidiomycetes possess a wide range of ligninolytic enzymes in their genome, including laccases, peroxidases and other oxidases (Figure 3A). Total auxiliary activity enzymes (AA) represent 19.8, 16.8 and 15.8% of the total annotated CAZymes in the genomes of Pbr985, Psan1264 and Lme1369, respectively [18,53]. AA enzymes are classified accordingly based on their ability to help carbohydrate-acting enzymes gain access to the plant sugars [54]. Proteomic analysis allowed the detection of a number of these annotated CAZymes in the secretomes of the studied fungi (Figure 3B, Spreadsheet S4).

Growth in the presence of technical lignin alone triggered the early secretion of many of these proteins, where in those cultures supplemented with maltose, the secreted oxidoreductases were only detected by day 7 of growth. This suggests that in the absence of an easily consumable carbon source, the fungi adjust their enzymatic machinery to be able to survive on the provided substrate, which in this instance is soda lignin. It has been previously shown that fungi are able to sense changes in their external environment and rapidly develop defense mechanisms to stress conditions [55]. Defense mechanisms include changes in the regulation of the expression of certain genes that are remarkably common between different strains and those include genes coding for proteins implicated in carbohydrate metabolism, protein metabolism and defense against reactive oxygen species. The results obtained here provide direct information on the induced proteins during growth on lignin, and therefore can be used to draw possible hypotheses on the different mechanisms involved in lignin modification by the studied fungi. The induced CAZymes at day 3 of growth on lignin alone belonged predominantly to the glycoside hydrolase (GH) and AA enzyme families for all three strains (Table 2).

Proteins belonging to the AA family were previously associated with lignin modification by either the oxidative generation of H_2_O_2_ or by directly attacking the lignin polymers in the presence or absence of mediators [56]. The action of these enzymes results in the oxidative cleavage of carbon–carbon and ether inter-unit bonds [54], consistent with the structural changes detected on lignin in this study and described below. The importance of H_2_O_2_-generating enzymes during growth on lignin is supported by their abundance in the analyzed secretomes of this study. Enzymes included glucose–methanol–choline oxidoreductases (AA3) and the copper radical oxidases (CRO, AA5_1). Psan1264 and Lme1369, showed a higher production of AA3s, specifically cellobiose dehydrogenase (AA3_1), aryl alcohol oxidase (AA3_2) and alcohol oxidase (AA3_3). On the other hand, Pbr985 secreted AA5_1 as the only detected H_2_O_2_-producing enzymes.

Based on the phylogenic analysis between secreted AA5_1 from Psan1264, Lme1369 and Pbr985, the identified AA5_1 proteins belonged to CRO1, CRO2, and CRO5 groups and were found to be induced on lignin for all strains (Figure 4). Within the AA5_1 family, CROs are classified under the subfamily of ‘’other copper radical oxidases’’ which includes five subfamilies with conserved catalytic residues compared to glyoxal oxidases, but for which the biological function is still unknown [57,58,59]. Interestingly, at least one protein belonging to each of these families was produced by the three fungi suggesting an important role of these specific proteins in lignin modification. Furthermore, Pbr985 and Lme1369 each produced an additional CRO2 protein highlighting the potential role of this isoform during growth on lignin. CROs were previously detected in secretomes during growth on lignin compounds, however, their role in lignin modification was never investigated [9,60]. Significantly upregulated CROs during the growth of *Phanerochaete chrysosporium* on lignin-rich substrates included CRO1 and CRO4 [60]. These enzymes have been shown to act on a broad substrate range including alcohol- and aldehyde-containing molecules [61,62] and their secretion during growth on lignin suggests a potential role in the oxidation of lignin functional groups.

Another interesting AA group of enzymes that was strongly present in lignin-containing conditions was LPMO. These enzymes were detected in the secretomes of Psan1264 and Lme1369. LPMOs are copper-containing proteins catalyzing the hydroxylation of C1 and/or C4 carbon in glycosidic bonds present on the surface of cellulose [63]. However, in the absence of more abundant levels of polysaccharides as substrate and the presence of electron donors, the enzyme promotes the uncoupled reduction of O_2_-generating H_2_O_2_ [64]. Electron donors include lignin-derived compounds released during biodegradation, diphenols produced by the reaction of different AA3, and by cellobiose dehydrogenases (AA8) [65]. A recent study has shown that H_2_O_2_ generated by LPMO was favorably used for lignin oxidation by the lignin-degrading peroxidases rather than for cellulose oxidation [66]. The induction of LPMO and their coupled secretion with AA3_1 support their role in lignin modification. However, lignin-degrading peroxidases were not detected in the secretomes of any of the studied strains. This leads to two hypotheses. The first being that although H_2_O_2_-generating enzymes and peroxidases have been previously shown to act synergistically during in vitro biomass conversion, their in vivo regulation varies considerably during growth and may also be dependent on the composition of the growth substrate (lignocellulose vs. lignin, for example). Peroxidase secretion was also previously found to be highly dependent on the growth conditions and was favored during solid state fermentation and in highly oxygenated cultures [67,68]. The second hypothesis could be that H_2_O_2_-generating enzymes can play a direct role in lignin modification in the absence of peroxidases. The role of H_2_O_2_ as a diffusible low molar mass oxidant in the initial non-enzymatic conversion of the plant biomass in brown rot fungi has widely been discussed (reviewed in [69]). Hydroxyl radicals formed by Fenton reaction were found to play a role in lignin modification in these organisms through demethylation/demethoxylation reactions [70].

In addition to GHs and AAs, a significant number of carbohydrate esterases (CE) were detected, particularly in the case of Lme1369. Two CE1 enzymes from Lme1369 and one from Psan1264 were secreted during growth on lignin alone. The protein sequences of the detected proteins were shared between 82 and 90% identity between each other. According to the CAZy classification system, the CE1 family contain most of the characterized fungal acetyl xylan esterases, feruloyl esterases and *p*-coumaroyl esterases [53,71]. These enzymes are involved in breaking down ester cross-links of lignin and hemicelluloses and in lignin solubilization [72,73]. Secreted CE1 enzymes in the current study might be involved in the release of the ferulic acid detected in the soluble fractions of fungal cultures, as described below. Other detected CE enzymes during growth on lignin (Lme1369 and Psan1264) included CE4, CE8, CE15 and CE16. Very few enzymes belonging to these families have been characterized and potential described activities include chitin deacetylase [74], 4-O-methyl-glucuronoyl methylesterase [75], and pectin methylesterase [76]. It is therefore difficult to describe the role that these enzymes play during growth on technical lignin. Interestingly, CE1 and CE4 were previously reported to be predominant in a lignin-degrading mixture of microorganisms found using a sugarcane soil sample as inoculum and lignin fragments as the major carbon source in minimal medium [77]. Similarly, in a recent study, fungal CE15 enzymes showed high activity on insoluble lignin-rich pellet from birchwood compared to smaller soluble lignin–carbohydrate complexes [75].

Several non-CAZyme and/or previously non-characterized proteins were also found to be induced on lignin (Appendix A). The intracellular nature of most of these proteins and their abundance in lignin cultures compared to sugar-containing controls could indicate cellular death and hyphal rupture, possibly caused by the lack of sugars in the media or by mechanical damage caused by insoluble lignin fragments in the shake flask conditions.

However, it is important not to exclude secreted proteins as the mechanisms employed by the fungi to survive and modify lignin can be very complex and involve more enzymes than the ones already described and characterized.

The proteomic analysis strongly suggested that the phenolic fraction of the technical lignin was modified by the fungi. To check this hypothesis and assess the possibility to use fungi to functionalize technical lignins, the composition of the soluble and insoluble fractions of Protobind 1000 and its extracts was investigated.

### 3.3. Analysis of Soluble Lignin Fraction after Fungal Treatment

Analysis of the lignin monomers and oligomers (i.e., compounds with a degree of polymerization (DP) below 12) extracted by EA from the culture supernatant showed that the incubation of the technical soda lignin with any of the three strains modified the composition of the water-soluble fraction of the material. High-pressure size exclusion chromatography (HPSEC) chromatograms (Figure 5) indicated an overall significant decrease in the amount of soluble compounds, with an almost total disappearance of all the higher molar mass compounds (eluted before 18 min) in the case of Psan1264. An increased proportion of monomers eluted at a retention time above 18 min in the case of Psan1264 and Pbr985 was observed. Monomers eluted between 17 and 19 min were then identified by LC–MS.

LC–MS analysis of the extracted supernatants confirmed that some phenolic compounds disappeared, at a rate depending both on the strain and the compound (Appendix A, Table 3).

Whereas ferulic acid disappeared in the presence of all strains, acetosyringone was metabolized only by Psan1264. Moreover, some common specific effects were observed with Psan1264 and Pbr985, such as the consumption of *p*-OH syringic acid and persistence of vanillin, which was not the case with Lme1369, where vanillin was consumed while syringic acid persisted. As observed by HPSEC, phenolic monomers abundance varied as follows: control > Pbr985 > Lme1369 > Psan1264. These observations support the substrate specificity of the three strains, where Psan1264 appears to be more active towards phenolic monomers.

The modifications observed in the lignin soluble fraction suggested that some compounds were possibly used as a carbon source by the fungi. However, their disappearance might also be due to their conversion into insoluble polymeric compounds through oxidation by the secreted oxidases. In all cases, the strains were found to have substrate specificities towards some phenolic monomers, e.g., acetosyringone consumption only observed with Psan1264. Acetosyringone is known as one of the most efficient natural mediators of laccase due to its favorable redox potential [78], allowing the oxidation of molecules with high redox-potential or a large size. When acting as a mediator, acetosyringone is regenerated through the oxidation process, thus its disappearance in Psan1264 cultures suggests that the laccases secreted by this strain would not use acetosyringone as a mediator in vivo.

### 3.4. Analysis of Insoluble Residual Lignin

Further evidence of the activity of the strains towards the phenolic substrate was ascertained through the analysis of the insoluble lignin fractions present in the centrifugation pellets. HPSEC indicated the consumption by all strains of almost all the phenolic monomers (eluted between 17 and 19 min) present in the culture pellets (Figure 6). In addition, a reduction in the relative proportion of lower molar mass oligomers (retention time between 14 and 16 min) was observed, specifically with Psan1264 and Pbr985, in parallel with an increase in the relative proportion of the higher molar mass oligomers and of the polymers (retention time between 11.5 and 14 min). This result confirms some similarity in the activity of Psan1264 and Pbr985 compared to Lme1369 and suggests that they were able to modify at least the lower molar mass compounds of the technical lignin insoluble fraction.

Structural analysis of this fraction (Table 4) revealed changes in terms of lignin functional groups, with a decrease in the total free phenol content observed during growth with Psan1264 and Pbr985. In the case of Psan1264, this modification was accompanied by a 40% decrease in the thioacidolysis yield. Since the thioacidolysis yield reflects the amount of lignin units only involved in β-O-4 linkages [28], the results suggest that the lignin has undergone structural changes, or at least changes in its composition. Due to the presence of mycelia in the centrifugation pellets, this decrease could partially result from an overall decrease in lignin proportion subsequent to the increase in mycelial mass.

## 4. Conclusions

In this work, we demonstrated that fungal growth on a technical soda lignin can be quantitatively measured. All the strains investigated led to the conversion of a portion of the phenolic monomers present in the technical soda lignin, with Psan1264 having the highest conversion rate. Psan1264 was also the only fungi that metabolized acetosyringone and was able to oxidize phenolic compounds of higher redox potential and molar mass. Pbr985 and Psan1264 showed similar specificities towards syringic acid and low molar mass oligomers, which were not converted by the other strains.

The secretomic profiles of these three strains were very different indicating that these fungi employ diverse enzymatic mechanisms in the utilization of lignins. The presence of certain enzymes in common between strains, especially copper radical oxidases, suggest that these enzymes are essential during lignin conversion and modification, thus allowing the fungi to grow in high lignified environments. In addition to these new insights into the wood-degrading fungi metabolism in the presence of technical lignins, this study offers perspectives for biorefinery. Indeed, it shows that lignin-rich recalcitrant biorefinery side streams could be modified and partially converted by fungi through the production of oxidases of potential industrial interest. Among these enzymes, H_2_O_2_-generating oxidases and LPMOs have been identified as promising ones for further mechanistic investigation and the future production of useful biocatalysts for the valorization of technical lignins.

## Figures and Tables

**Figure 1 jof-07-00039-f001:**
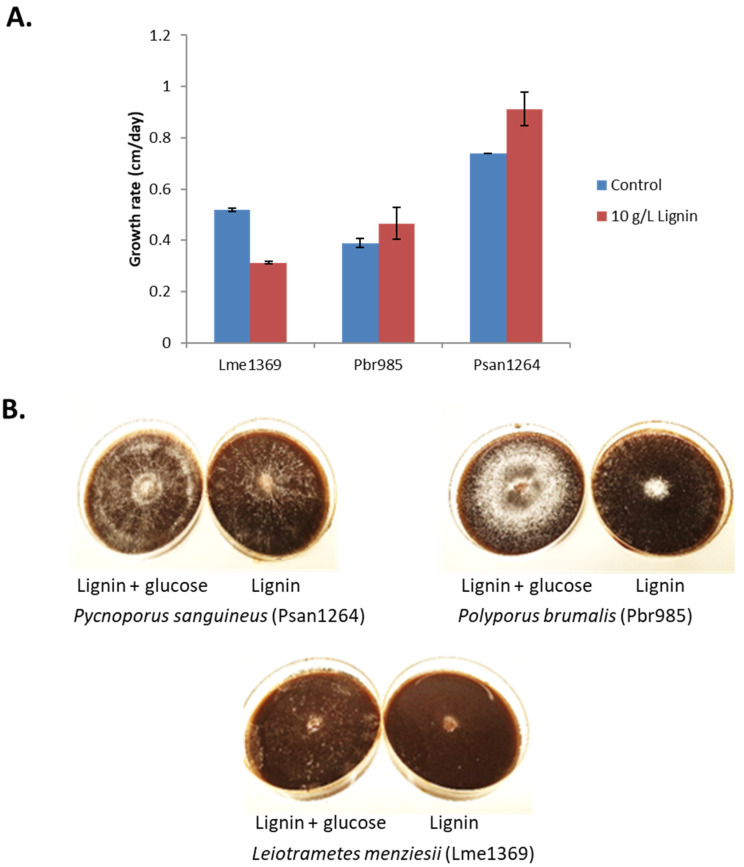
Fungal growth on lignin-containing plates. (**A**) Growth rate expressed as diameter growth per day on minimal media plates containing 10 g·L^−1^ technical soda lignin. Control plates were supplemented with 1 g·L^−1^ glucose. Values are the means and bars which indicate standard deviations. (**B**) Observed mycelial density during growth on Lignin after 14 days in the presence and absence of glucose for Psan1264, Pbr985 and Lme1369.

**Figure 2 jof-07-00039-f002:**
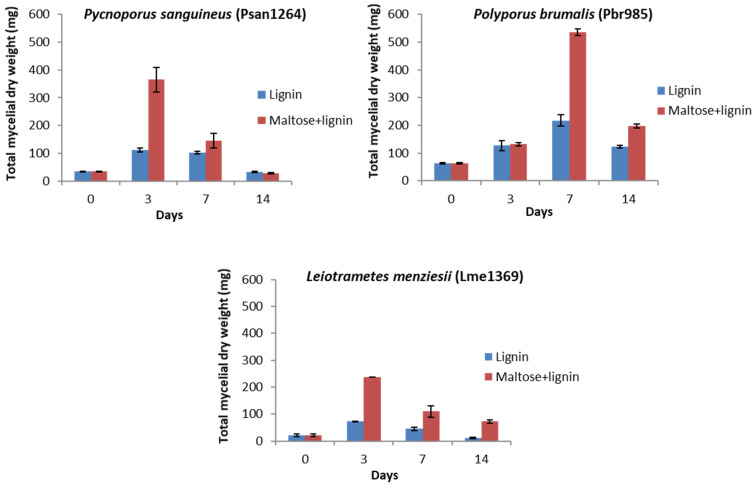
Mycelial dry weight determination in liquid cultures. Determined mycelial dry weights for Psan1264, Pbr985 and Lme1369 in control (maltose 2.5 g·L^−1^ + lignin 15 g·L^−1^) and 15 g·L^−1^ lignin conditions at days 0, 3, 7 and 14 after inoculation. Values are the means of three samples and bars representing the standard deviations.

**Figure 3 jof-07-00039-f003:**
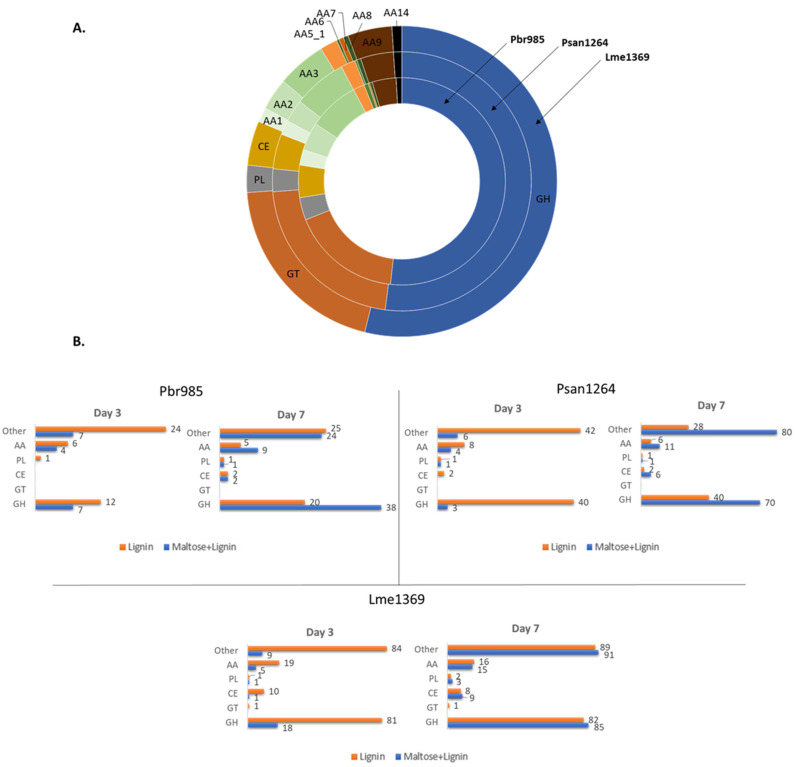
Distribution of predicted CAZymes in the genomes and secretomes of the studied fungi: (**A**) distribution of CAZymes in the genomes of Pbr985 (inner ring), Psan1264 (middle ring) and Lme1369 (outer ring); (**B**) number of different CAZymes detected in the secretomes in lignin and in maltose + lignin conditions at days 3 and 7 of growth for the three tested strains. AA, auxiliary activity enzymes; GH, glycoside hydrolases; GT, glycosyl transferases; PL, polysaccharide lyases; CE, carbohydrate esterases; AA1, multicopper oxidases; AA2, class II lignin-modifying peroxidases; AA3, glucose-methanol-choline (GMC) oxidoreductases; AA5_1, copper radical oxidases; AA6, 1,4-benzoquinone reductase; AA7, glucooligosaccharide oxidases; AA9, AA11 and AA14 lytic polysaccharide monooxygenases; AA12, pyrroloquinoline quinone-dependent oxidoreductase.

**Figure 4 jof-07-00039-f004:**
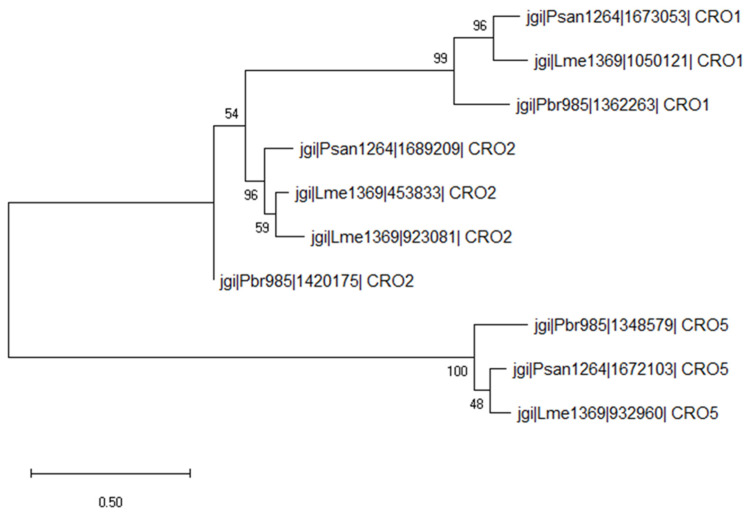
Evolutionary analysis by the maximum likelihood method of secreted AA5_1 proteins. The evolutionary history was inferred by using the maximum likelihood method and Poisson correction model. The tree with the highest log likelihood (−12,807.31) is shown. The percentage of trees in which the associated taxa clustered together is shown next to the branches. Initial tree(s) for the heuristic search were obtained automatically by applying the neighbor-join and BioNJ algorithms to a matrix of pairwise distances estimated using a JTT model, and then selecting the topology with the superior log likelihood value. Evolutionary analyses were conducted in MEGA X [36].

**Figure 5 jof-07-00039-f005:**
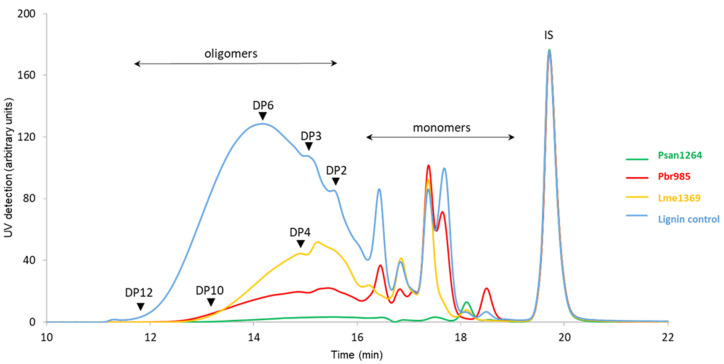
HPSEC analysis of water-soluble residual lignin fraction: high-performance size-exclusion chromatograms of the ethyl acetate extracts of the culture supernatants recovered from soda lignin 14-day incubation with Pbr985, Psan1264, or Lme1369; chromatograms normalized on IS. Eluent tetrahydrofuran (THF), 1 mL·min^−1^; detection at 280 nm; 100 Å PL-gel column (Polymer Laboratories, 5 μm, 600 mm × 7.5 mm). DP: degree of polymerization, IS: internal standard is toluene.

**Figure 6 jof-07-00039-f006:**
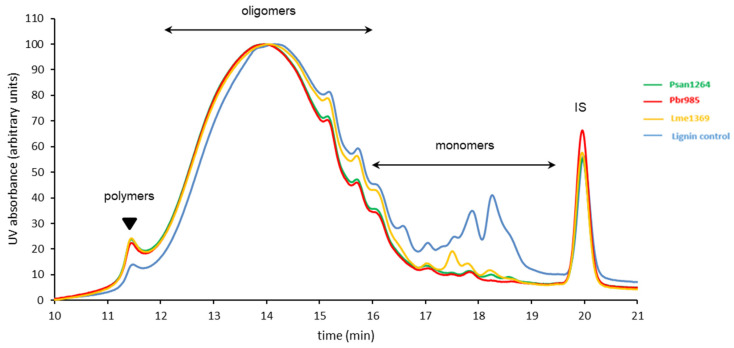
HPSEC analysis of water-insoluble residual lignin fraction. High-performance size-exclusion chromatograms of the water-insoluble residues recovered from soda lignin after 14-day incubation with Pbr985, Psan1264, or Lme1269. The lignin control was dissolved in the culture media in the absence of fungi and incubated under the same conditions. Eluent THF, 1 mL·min^−1^; detection at 280 nm; 100 Å PL-gel column (Polymer Laboratories, 5 μm, 600 mm × 7.5 mm). IS: internal standard.

**Table 1 jof-07-00039-t001:** ABTS and veratryl alcohol oxidation tests. The ability of the different culture supernatants to oxidize ABTS and veratryl alcohol was measured. The presented data are the mean values (±standard error) and the darker green color highlights higher detected activities. nd: activity not detected.

Enzyme	Strain	Condition	Incubation Time (Day)		
			3	7	14
			Activity (nkat/mg dry mycelia)		
ABTS Oxidation	Pbr985	Maltose (20 g/L)	1.47 ± 0.09	26.61 ± 8.05	7.42 ± 0.33
		Maltose (2.5 g/L) + Lignin (15 g/L)	4.22 ± 0.29	34.94 ± 2.57	226.42 ± 32.72
		Lignin (15 g/L)	0.49 ± 0.24	14.34 ± 1.89	37.36 ± 5.59
	Psan1264	Maltose (20 g/L)	35.56 ± 1.24	32.18 ± 2.79	25.38 ± 0.74
		Maltose (2.5 g/L) + Lignin (15 g/L)	20.74 ± 1.61	768.67 ± 151.03	3175.82 ± 320.50
		Lignin (15 g/L)	13.97 ± 0.12	32.52 ± 3.26	112.49 ± 13.09
	Lme1369	Maltose (20 g/L)	nd	nd	nd
		Maltose (2.5 g/L) + Lignin (15 g/L)	34.16 ± 2.49	66.18 ± 0.93	73.08 ± 5.62
		Lignin (15 g/L)	nd	nd	6.52 ± 1.31
Veratryl Alcohol Oxidation	Pbr985	Maltose (20 g/L)	nd	nd	0.11 ± 0.15
		Maltose (2.5 g/L) + Lignin (15 g/L)	0.97 ± 0.06	1.19 ± 0.09	2.40 ± 0.1
		Lignin (15 g/L)	nd	1.91 ± 0.37	nd
	Psan1264	Maltose (20 g/L)	nd	nd	nd
		Maltose (2.5 g/L) + Lignin (15 g/L)	nd	0.25 ± 0.19	0.36 ± 0.1
		Lignin (15 g/L)	nd	2.27 ± 0.52	1.97 ± 0.1
	Lme1369	Maltose (20 g/L)	nd	nd	nd
		Maltose (2.5 g/L) + Lignin (15 g/L)	nd	0.38 ± 0.04	1.94 ± 0.28
		Lignin (15 g/L)	nd	nd	0.09 ± 0.1

**Table 2 jof-07-00039-t002:** The auxiliary activity enzymes and carbohydrate esterases identified in the secretomes. Highlighted in green are the proteins that were induced at day 3 of growth on lignin alone and the stars indicate proteins that were exclusively detected in the absence of maltose. JGI IDs are used to represent the enzymes (https://genome.jgi.doe.gov/portal/).

Pbr985			Psan1264			Lme1369		
Protein ID	CAZy	Day of Growth	Protein ID	CAZy	Day of Growth	Protein ID	CAZy	Day of Growth
657909	AA1_1	3 and 7	1583166	AA1_1	3, 7 and 14	924031	AA1_1	3 and 7
224137	AA1_1	3, 7 and 14	1560767	AA1_1	3	1044554	AA1_1	3, 7 and 14
1420175	AA5_1	3, 7 and 14	1672751	AA8-AA3_1	3, 7 and 14	916190	AA8-AA3_1	3, 7 and 14
1362263	AA5_1	3 and 7	1574363	AA3_2	3, 7 and 14	927121	AA3_2	3 and 7
1348579	AA5_1	3	1648421	AA3_4	7 and 14	924050 *	AA3_2	3
1388977	AA7	3, 7 and 14	1672103	AA5_1	3, 7 and 14	1048304	AA3_2	7
			1689209	AA5_1	3, 7 and 14	437597 *	AA3_3	3, 7 and 14
			1673053 *	AA5_1	3	453833	AA5_1	3, 7 and 14
			1600019 *	AA9	3	1050121 *	AA5_1	3 and 7
			1677933 *	CBM1-CE1	3	923081	AA5_1	3, 7 and 14
			1565370	CE4	3 and 7	932960	AA5_1	3 and 7
			1577721	CE8	7 and 14	1057183 *	AA9	3 and 7
						1050920	AA9	3 and 7
						1050025	AA9	3 and 7
						497583	AA9	3
						95318	AA9	3, 7 and 14
						1046503 *	AA9	3
						923917 *	AA9	3
						931673	AA9-CBM1	3 and 7
						1106760 *	AA9-CBM1	3
						932017 *	AA14	7
						1042341	CE1-CBM1	3 and 7
						693655 *	CE1-CBM1	3
						946461	CE4	3 and 7
						908440	CE4	3 and 7
						754929 *	CE4	3
						1018647	CE8	3, 7 and 14
						46074	CE8	7 and 14
						921049	CE15-CBM1	3
						250112	CE16	3 and 7
						1049410	CE16	3 and 7
						1046069	CE16-CBM1	3 and 7

**Table 3 jof-07-00039-t003:** Phenolic compounds observed by LC–MS. LC–MS of the ethyl acetate extracts of the culture supernatants after growth of Lme1369, Psan1264 or Pbr98 on lignin alone compared to the lignin control (lignin dissolved in the culture media in the absence of fungi and incubated under the same conditions): ~almost constant, -significant decrease, --almost disappeared.

Rt (min)	Phenolic Compounds	Lme1369	Psan1264	Pbr985
1.9	*p*-OH benzaldehyde	--	-	~
2.3	Syringic acid	~	--	--
2.8	Vanillin	--	-	-
3.4	*p*-Coumaric acid	--	--	-
3.8	Syringaldehyde	--	--	~
4.7	Ferulic acid	--	--	--
5.0	Acetosytingone	~	--	~

**Table 4 jof-07-00039-t004:** Structural analysis of the water-insoluble residues recovered from lignin after fungal treatment. PB1000 refers to the substrate before any treatment. The lignin control was dissolved in the culture media in the absence of fungi and incubated under the same conditions.

Strain	PheOH (mmol·g^−1^) ^a^	Thioacidolysis ^b^	
		Total Yield (µmol·g^−1^)	S/G Ratio
PB1000	2.68	104 ± 4	0.97 ± 0.01
Control	2.90	132 ± 9	0.93 ± 0.05
Pbr985	2.42	96 ± 1	0.91 ± 0.02
Psan1264	2.48	79 ± 7	0.85 ± 0.02
Lme1369	2.85	120 ± 4	0.90 ± 0.04

^a^ Data correspond to single determination ^b^ Data are the mean values (±standard error) between duplicate analyses.

## Data Availability

The data presented in this study are available on request from the corresponding author. The data are not publicly available due to confidentiality.

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
