# Peer review of "Fungal Treatment for the Valorization of Technical Soda Lignin"

_jof, 2021, doi:10.3390/jof7010039_

Round 1
Reviewer 1 Report
The authors described an interesting work on soda lignin degradation. In general the article is sound in results and writing. However, the authors need to go through this article carefully as there are a few wrting errors throughout the article. for example, line 271: contentslower---. figure legends need to describe clearly and in detail. Figure 1, what is the control? Figure 2, the setting for the vertical coordinate is not appropriate. Table 3, what is lignin control? some enzyme names? it is totally confusing!
Author Response
Reviewer 1:
- The authors described an interesting work on soda lignin degradation. In general the article is sound in results and writing. However, the authors need to go through this article carefully as there are a few wrting errors throughout the article.
Thank you for your feedback. The authors went through the manuscript carefully and corrected the writing errors. In particular, American English spelling was used consistently all along the manuscript. Moreover, some uncorrect formulation were detected and corrected.
- Line 98: “The soda technical lignin was produced from a wheat straw and Sarkanda grass mix (Protobind 1000) and purchased from GreenValue Enterprises LLC (Media, PA, US). The sample was comprised of Klason lignin (88.1 %), carbohydrates (1.9 %, of which 1.2 % xylose, 0.3 % arabinose, 0.1 % galactose and 0.2 % glucose), free phenolic monomers (1.4 %) and ash (1.4 %).” changed for “The soda technical lignin (Protobind 1000) was produced from a wheat straw and Sarkanda grass mix and purchased from GreenValue Enterprises LLC (Media, PA, US). The sample was analyzed for Klason lignin (88.1 %), carbohydrates (1.9 %, of which 1.2 % xylose, 0.3 % arabinose, 0.1 % galactose and 0.2 % glucose), free phenolic monomers (1.4 %) and ash (1.4 %) contents.”Line 210: “polymerisation” changed for “polymerization”
- Line 238: “derivatisation” changed for “derivatization”
- Line 448: “low molecular weight” changed for “low-molar mass”
- Line 507; 509: “monophenolic compounds” changed for “phenolic monomers’
- Line 506: “As observed by HPSEC, monophenolic compounds abundance followed: control > Psan1284 > Lme1369 > Pbr985.” changed for “As observed by HPSEC, phenolic monomers abundance varied as follows: control > Pbr985 > Lme1369 > Psan1264.”
- line 271: contentslower---.
This error was corrected.
- figure legends need to describe clearly and in detail.
More information and details were added to the figure lengends.
- Figure 1, what is the control?
Control plates were contained 1 g.L-1 glucose in addition to 10 g.L-1 technical soda lignin. This information was added to the figure legend.
- Figure 2, the setting for the vertical coordinate is not appropriate.
The vertical coordinate was adjusted to a maximum of 600 mg.
- Table 3, what is lignin control? some enzyme names? it is totally confusing!
This table represents the phenolic compounds detected by LC-MS in the culture supernatants (soluble fractions). The supernatants were collected after 14 days of growth of Lme1369, Psan1264 or Pbr98 on lignin alone and the abundance of each phenolic compound was compared to the control condition. The control consisted of the lignin substrate dissolved in the culture media in the absence of fungi and incubated under the same conditions. These details were added to the table legend and the table was corrected.
Reviewer 2 Report
Several minor comments and suggestions:
- Figures 1,2,3 are somewhat unclear and dazed. Need to use improved graphics
- Figure 2: X axis should be changed to 600 maximum, instead of 1200. This will improve the visibility of the results.
- Table 3: I suggest to complete the border of each treatment from the name of the fungus to the end of the line to distinguish between the results of each fungus.
- The same word at line 366 and 365 is written differently, once CAZymes and once Cazymes . Needs uniformity in writing. Please to check all the MS.
- Table 2- is to long with too much details. Is it possible to shorten by summarizing it or to add it as supplementary.
- Line 548- Which 2 fungi? The MS is dealing with 3 fungi.
Good Luck!
Author Response
- Figures 1,2,3 are somewhat unclear and dazed. Need to use improved graphics
The quality of the figures was enhanced and the text size in the figures was increased to improve reading.
- Figure 2: X axis should be changed to 600 maximum, instead of 1200. This will improve the visibility of the results.
The maximum of the Y axis was changed to 600 mg instead of 1200 mg.
- Table 3: I suggest to complete the border of each treatment from the name of the fungus to the end of the line to distinguish between the results of each fungus.
The borders to separate the three strains were added.
- The same word at line 366 and 365 is written differently, once CAZymes and once Cazymes . Needs uniformity in writing. Please to check all the MS.
The word was corrected and written the same way in the manuscript.
- Table 2- is to long with too much details. Is it possible to shorten by summarizing it or to add it as supplementary.
The table was adjusted to include auxiliary activity enzymes and carbohydrate esterases only. The full secteromes are available as supplementary.
- Line 548- Which 2 fungi? The MS is dealing with 3 fungi.
This error was corrected.